# Effect of Contact Conformity on Grease Lubrication

**Michal Okal \*** , **David Kostal \***, **Petr Sperka** , **Ivan Krupka and Martin Hartl**

Faculty of Mechanical Engineering, Brno University of Technology, Technicka 2896/2,
616 69 Brno, Czech Republic

\* Correspondence: michal.okal@vut.cz (M.O.); kostal@fme.vutbr.cz (D.K.)

**Abstract:** This work focused on the experimental study of grease lubrication mechanisms around contacts in radial ball bearing 6314. The main objective of this work was to show the influence of conformities and their impact on grease lubrication in ball bearings. For the experiments, a tribometer of ball-on-ring configuration was used and fluorescence microscopy was chosen as the observation method. The results showed that, under starvation conditions, a conformity of 0.52 at velocities around 2 m/s produces a 50% thicker lubricating film than a conformity of 0.58. The available amount of lubricant around the contact area for conformity 0.52 was approximately three times less than that for conformity 0.58, and the same ratio was observed for the amount of lubricant on the rolling track. Experiments show that a realistic tribometer geometry allows a more accurate experimental study of the lubrication mechanisms of greases in ball bearings.

**Keywords:** grease lubrication; ball bearing; starvation; replenishment

## 1. Introduction

At present, 90% of rolling bearings are lubricated with grease. These are semi-fluid mixtures consisting of a base oil, thickener and additives [1] and are one of the main components of bearings, with grease lubrication parameters strongly influencing bearing life. They have the character of Bingham liquids, i.e., flow occurs only after a certain shear stress is exceeded. This characteristic makes them preferable to oils in applications where lubrication is required without continuous refills. However, their disadvantage is their temporary lifetime, which we cannot predict accurately. There is a synergy between bearing design and grease properties. Different bearing types require different greases, and bearing and grease parameters interact. Lubrication processes in the ball bearing can be divided into several basic phases. The initial phase of bearing operation is called the churning phase [2]. During this phase, the lubricant is redistributed throughout the bearing, with the majority of the lubricant remaining attached to the cage. The phase is accompanied by higher temperatures and higher resistance due to the rolling elements wading in the lubricant. Following the churning phase, the clearing phase occurs when a limited amount of grease remains in the rolling raceways. After the initial phases, the bleeding phase occurs, which prevails for the rest of the bearing life. The lubrication is mainly due to the "bleed" base oil [3]. Bleeding is a gradual process where only a certain amount of oil is released. This amount is reduced during bearing operation due to lubricant evaporation and leakage from the bearing. Furthermore, the lubricant is degraded due to oxidation and other degradation processes. Therefore, during the bleeding phase, moments of severe starvation occur when a sufficiently thick lubricating layer is not formed in the contacts. The temperature rises due to friction, however, will help to facilitate the release of new base oil, which will again return sufficient lubricant to form a lubricating film. The processes of heavy starvation and lubricant replenishment are stochastic in nature and occur throughout the bleeding phase. Rolling bearings operate under starvation conditions for most of their operating life, so it is essential to have a good understanding of how greases work under these conditions in rolling bearings.

The amount of lubricant available in the surrounding area, especially before contact, indicates how well lubricated the contact is. The lubricant–air interface before the contact is referred to as the inlet meniscus, and the position of this meniscus is one of the determining parameters for the level of starvation. A limited amount of lubricant before contact will cause a faster pressure rise compared to fully flooded conditions where the onset of the rise is at an infinitely distant point [4,5]. The limited pressure rise limit will result in insufficient lubricant film formation at the contact. Wedeven was the first to observe EHD starvation of the contact and a decrease in film thickness in the contact with decreasing meniscus distance [6]. Owing to the rolling element crossing, the lubricant tends to spread into two side bands from which the lubricant is dosed into the contact area [7]. The process where the lubricant is refilled into the contact is defined as replenishment, and Chiu specified a formula to determine the amount of lubricant to be refilled into the rolling path after passage [8]. The resulting starvation level is then determined as a balance between the loss mechanisms and the lubricant replenishment processes in the contact [9]. The key parameters are lubricant viscosity, velocity, surface tension, capillary force, and available lubricant quantity. Chevalier stopped using the distance of the inlet meniscus and started using the amount of lubricant at the pre-contact boundary [10,11]. This approach was much more suitable and accurate for determining the starvation level and developed a theoretical model for predicting the central thickness as a function of the amount of lubricant before contact.

The distribution of lubricant in the bearing is essential to determine the starvation level. Van Zoelen et al. developed a model to determine the migration of the lubrication layer due to centrifugal forces [12,13]. Using the model, they were able to simulate the film drop due to overrides, but they did not include lubricant replenishment processes in the model. Furthermore, the model was extended to include Van der Waals forces [14]. The observed starvation processes were divided into three stages where, for moderate levels of starvation, the sidebands are still connected before contact, due to capillary forces. Subsequently meniscus rupture occurs, and surface tension plays a key role in this case. As the velocity is further increased, the film thickness decreases to an almost stable value of about 20 nm. This thin layer of lubricant is generated precisely due to Van der Waals forces. These forces are intense near the surfaces of the bodies and thus provide an effective mechanism for lubricant replenishment. This condition is referred to as parched lubrication [15]. The numerical model of lubricant replenishment in contact has been further discussed in references [16,17]. Cen et al. confirmed that replenishment occurs predominantly before contact, when he observed the film thickness in a real bearing with a different number of rolling elements [18]. The lower number of elements caused a decrease in film thickness instead of the expected increase, so the increase in time between passes was not favourable. The effect of centrifugal forces was observed in reference [19], where the changes in side lubricant reservoirs were recorded through a high speed tribometer. Contact was made on the ball-on-disc tribometer and, at high speeds, the reservoir further away from the axis of rotation of the disc disappeared. As a result, the film thickness was not uniformly formed in the contact, but the part with the smaller side band showed a more pronounced degree of starvation. The effect of centrifugal forces and the different contact geometry of the ball-on-disc tribometer cause different lubricant replenishment around the contact, which then affects the resulting central film thickness.

Works describing lubrication mechanisms with grease often rely on the ball-on-disc tribometer. Cann observed a higher lubricant film thickness at initial speeds, which gradually decreased with increasing disk speed [20]. The reason for the greater thickness is the thickener particles that come into contact. The thickener particles tend to stick to the contact surfaces and form a film thickness that can be divided into two components [21], the part formed by thickener and the part formed by oil, which can be determined by EHL theory. After a long period of time, the film thickness decreases and stabilizes. The stabilization is aided by the bleeding of oil from the grease structure that flows into the rolling path [22]. The degree of starvation then mainly depends on the amount of lubricant available, the

willingness of the grease to release the base oil, viscosity, and speed. In the case of grease starvation conditions, theoretical calculations are unable to predict the thickness of the lubricating film. Therefore, experimental methods are used in these cases.

Although much effort has been devoted to understanding the starvation phenomenon, there is still a lack of linking of the basic knowledge from ball disc tribometers to actual bearings. This is due to the unreliable geometry of the ball-on-disc tribometer and the lack of transparency of the bearing ring, which prevents direct observation of the lubrication mechanisms. Another reason is the complex rheological properties of the grease which cause unpredictable behaviour. This study relies on the combination of realistic bearing geometry and single element transparency, which allows the use of optical methods. This coupling allows the experimental study of starvation arising directly in radial ball bearings. More specifically, this study focuses on the effect of conformity on the sensitivity of contacts to starvation.

## 2. Materials and Methods

The experiments were performed on a tribometer with a ball-on-ring configuration shown in Figure 1. The core of this test rig is a ring made of borosilicate glass BK7 with a geometry taken from a deep-groove ball bearing (6314). The glass ring has an internal groove with a diameter of 26.4 mm. A thin lubricating film is formed due to the relative motion of the glass ring and the AISI 52100 (100Cr6) bearing steel ball of different diameters (maximum 25.4 mm). The experimental device operates under a pure rolling assumption. To verify this assumption, control tests were performed. The rolling element is supported by two shafts which are mounted in bearings. The rolling element is supported so that the reaction forces are directed to the centre and the formation of the track on the rolling element is not disturbed. The glass ring is driven by a servo motor and the rolling element is pressed against the glass ring by a lever system. The optical imaging system consists of an industrial microscope, a digital camera (sCMOS), a mercury lamp, and a computer.

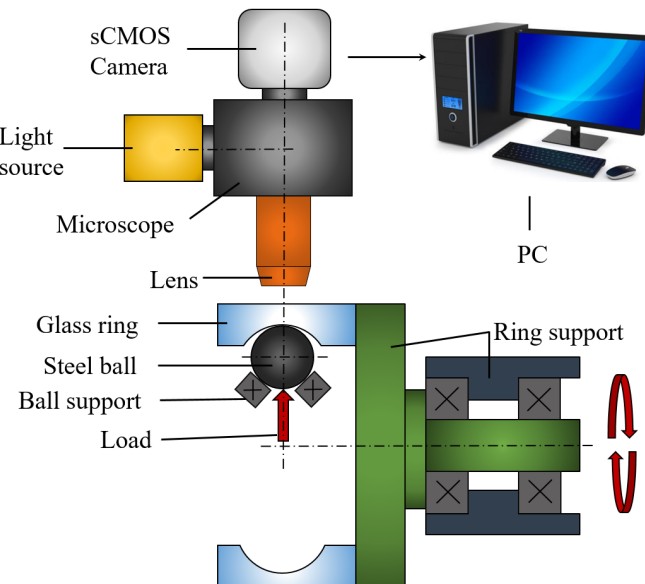

**Figure 1.** Scheme experimental of apparatus.

The optical method used for the film thickness evaluation and the contact surrounding observation was fluorescence microscopy. This method was chosen because of its ability to analyze directly the amount of lubricant, so that it can be used to analyze the distribution of lubricant around the contact, as well as to directly measure the film thickness in the contact. A mercury lamp was used to illuminate the sample. To excite the sample in the simulator, a specific excitation filter was used to isolate the desired wavelengths. The excited lubricant emits light of slightly longer wavelength, due to Stoke shift, which is transmitted through

a dichroic mirror and isolated by the emission filter. Only this light is then captured by a 16-bit monochrome sCMOS camera. The principle of this method is described in three following steps [12]:

- Excitation: Photon which is emitted from an external source (mercury lamp, laser), and is absorbed by fluorophore and creates excitation.
- Excited-state lifetime: Takes 1–10 ns. Molecules can undergo energy dissipation and are left in a state which can emit fluorescence.
- Fluorescent emission: A photon of energy is emitted and fluorophore returns to its ground state. Owing to energy dissipation during the previous phase, the photon has a longer wavelength than the excited photon.

Proper calibration is crucial to correctly evaluate film thickness. The sandwich setup was used for calibration as shown in Figure 2. Calibration images are obtained from the capture of the gap with known contact thickness and obtained intensity is assigned. The calibration has been described in more detail by Kostal [13].

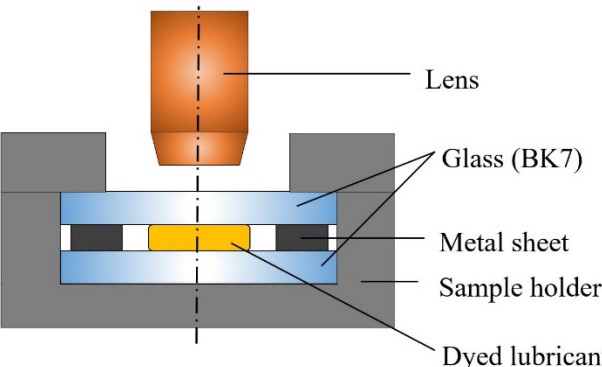

**Figure 2.** Scheme of the calibration setup.

### 2.1. Lubricants

A lithium complex-based grease was used for the experiments. A fluorescent dye was added to the base oil in the grease. Some experiments were performed with a base oil with identical viscosity to the base oil used in the grease to validate the results. A fluorescent dye was also added to the base oil at the same concentration. Further details of the lubricants are shown in Table 1.

**Table 1.** Properties of used grease.

| Grease Property | Grease 1 | Mineral Base Oil |
|---|---|---|
| Recommended operating temperature [°C] | −30 to +120 | −30 to +120 |
| NLGI Grade [−] | 2–3 | - |
| Base oil kinematic viscosity at 40 °C [mm$^2$ s$^{-1}$] | 50 | 50 |
| Penetration at 25 °C (ISO 2137) [10–1 mm] | 270 | - |
| Thickener | Li−soap | - |

### 2.2. Fluorescent Dye

Pyrene was used as a fluorescent dye. The sample enrichment process was carried out first for the base oil. Subsequently, the oil was then used for the formation of the grease. With this approach, only the base oil was dyed. The fluorescent dye content was 1 mass %.

### 2.3. Species

Experiments were carried out with rolling elements of different diameters. Different element diameters produce different sizes of contact areas and conformities. The size of the

chosen diameters produces a range of conformities that is standard for ball rolling bearings. Further details of the elements are listed in Table 2.

**Table 2.** Specifications of rolling elements.

| Ball Diameter [mm] | Conformity (f) [−] | Ellipticity of Contact [−] |
|---|---|---|
| 25.4 | 0.52 | 8.4 |
| 23.8 | 0.55 | 4.5 |
| 22.2 | 0.58 | 3.3 |

*2.4. Conditions*

All experiments were performed with a contact pressure of 0.3 GPa. The temperature during the experiment was around 26 ± 0.5 °C. The range of velocities during the experiments was 0–2 m/s. Two regions were observed during the experiments.

The observed area (1) gives direct information about the amount of lubricant in the contact area. The location of the observed area was directly in the area of contact. For all three conformities, this area was the same size (0.5 × 0.2 mm) during the measurement. The observed area (2) gives information about the amount of lubricant available in the immediate vicinity of the contact. The region of interest (ROI) was rectangular in shape, with the width corresponding to twice the width of the contact and the height was always 1.6 mm. The location of the ROI was as shown in Figure 3, i.e., the bottom of the rectangle was ploughed with the bottom of the contact area.

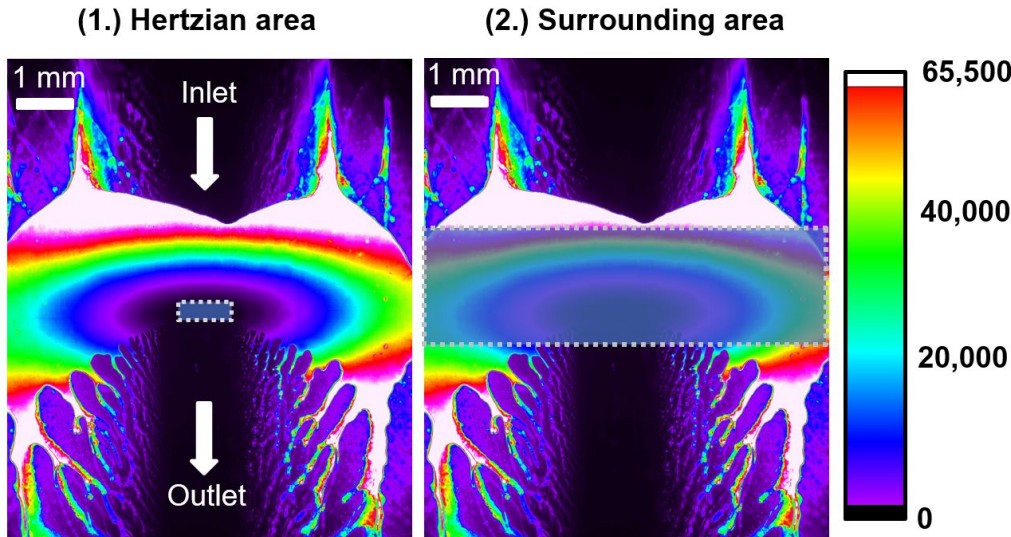

**Figure 3.** Observed areas.

An exposure time of 0.4 s was set during the contact intensity measurements. The data for each measurement were measured three times and the values used were averaged among themselves. The recording of fluorescence intensity from the central contact area was only done at specific velocities. The recording was always done at regular intervals where, after setting the desired speed, the recording was done after 10 s. This dwell time before recording ensured an observable effect of velocity. Each measurement was performed in an identical manner, for all conformities, to promote repeatability.

For experiments with base oil, a micropipette was used to enable precise dosing of the lubricant. The dosing was done gradually and into a groove in the ring. For the oil experiments, 50 µL was dosed in five doses. During the application process, the rolling element was in contact with the ring and the ring was rotated. The rotation speed was set at 5 rpm. Due to the rotation of the ring and the gradual dosage, the lubricant was evenly applied to the groove. During each new experiment, the lubricant application

procedure was identical. For the grease experiments, a quantity of 0.5 g of grease was used. The quantity was always weighed and applied to the rolling element. During the application of the grease, the ball was pressed against the ring and it was also rotated at 5 rpm. The application was gradual, where over one revolution the entire amount of grease was applied to the groove in the ring.

## 3. Results and Discussion

### 3.1. Central Film Thickness at a Range of Speeds

The following charts in Figure 4 show the trend during measurements with different conformities. The values for oil (left) and for grease (right) are shown and the trends are very similar. There are three curves corresponding to a given conformity, and a curve representing the fully flooded conditions averaged over all three conformities in each chart.

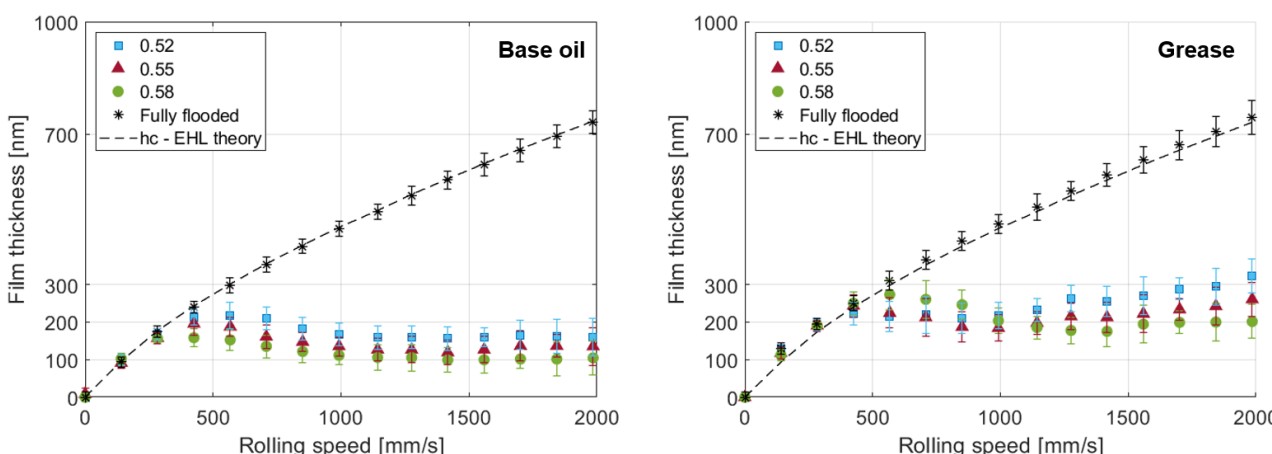

**Figure 4.** Effect of conformity on central film thickness (**left**—oil, **right**—grease).

There are, in Figure 4, two graphs which show the film thickness when using base oil and the graph on the right shows the film thickness when using grease. There are three curves in the graph representing the evolution of the central thickness on velocity, according to the chosen conformities, and a curve showing fully flooded conditions. The fifth curve represents the theoretical values of the central thickness as a function of viscosity and entrainment velocity ($\eta u$). The transition from fully flooded conditions occurred at around 350 mm/s in the case of oil. The difference between the conformities can be seen, with the smaller conformity showing a later transition and a higher film thickness value. However, all three curves have a similar shape. There was a gradual decrease in thickness from the velocity of 350–1000 mm/s. From a velocity of 1000 the evolution was minimal, and the values remained around 100–160 nm depending on the conformities.

In the case of grease lubrication, one can see the difference between the comparison with oil and between the individual conformities. Compared to oil, there was a deviation from the theoretical values at a higher speed of 450 mm/s. At lower speeds, higher experimental values than the theoretical ones can be observed. These are followed by a slight decrease, up to a velocity of 1000 mm/s, and then a gradual increase in thickness from 1000–2000 mm/s. Comparing the conformities, an earlier transition from the theoretical values can be observed for the 0.52 conformity and a later one for the 0.58 conformity. At 1000 mm/s, the conformities show similar film thickness values around 200 nm. The gradual increase in film thickness from this velocity varied with the conformities, with the conformity 0.52 showing a higher thickness than the conformity 0.58. At 2000 mm/s, the thickness ranged from 200–310 nm depending on the conformities. When comparing the theoretical curve and the fully flooded conditions, a match can be seen in the case of lubrication with oil.

To assess the level of starvation in selected conformities, a ratio between hc (central film thickness) and hcff (fully flooded central film thickness) was created. How this ratio

varied as a function of speed can be seen in Figure 5. The graph only describes the values for grease lubrication. A gradual decrease in values, in the range 0–1000 mm/s, can be seen and then a stabilization in the range 1400–2000 mm/s. In the initial part, the curves overlap, but subsequently, after stabilization, a visible difference between the conformities emerges. Conformity 0.52 stabilized at around 0.42, while conformity 0.58 stabilized at around 0.3.

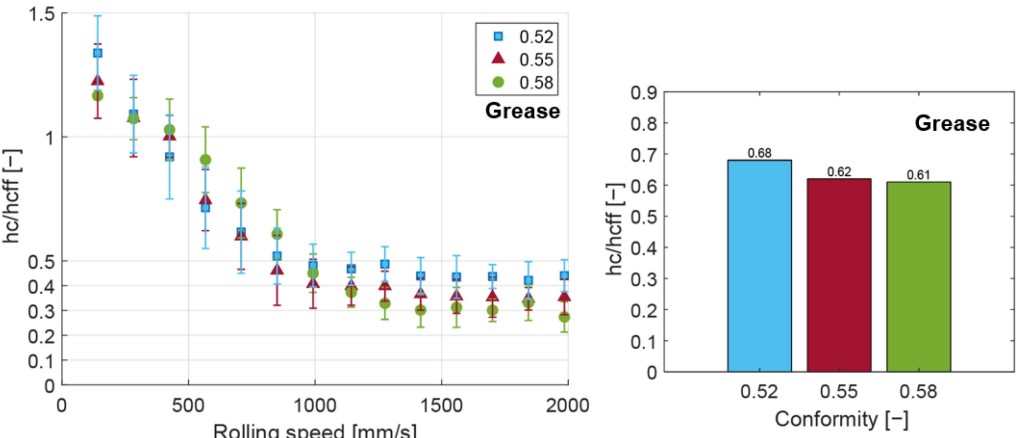

**Figure 5.** hc/hcff ratio (**left**—influence of speed, **right** average value).

Figure 5 on the right shows the averaged values of the whole curves for the individual conformities. Averaging the values over the entire velocity range reduces the differences between conformities but gives one value describing the starvation resistance of a given conformity. Although the 0.52 conformity had a slightly lower ratio than the other conformities at low speeds, it had the highest overall average. The remaining conformities, 0.52 and 0.58, showed very similar values.

Figure 6 shows images of the contact with the selected conformities at different velocities. It can be observed that the 0.52 conformity produced a wider contact and the side reservoirs are further away from the bottom of the groove and, according to the lower fluorescence intensity, there is less lubricant. In the case of conformity 0.58, the lateral reservoirs were closer to the center of the groove bottom and showed more lubricant. At lower velocities, the meniscus before contact was also larger in the 0.58 case than in the 0.52 case. At higher velocities, the meniscus ruptures and the shape of the lateral reservoirs hardly changes due to velocity.

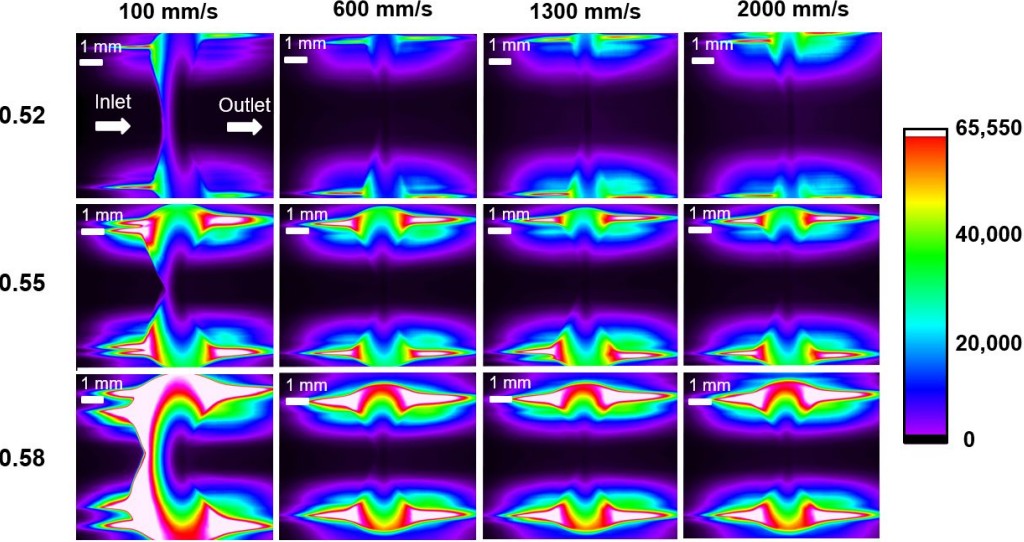

**Figure 6.** The distribution of the grease around the contact at different speeds.

The results of the ball-on-ring device show considerable similarity to those of the ball-on-disc device [23,24]. The difference occurs mainly during starvation, when there is not such a significant decrease in thickness. This difference is mainly due to centrifugal forces and greater capillarity. When comparing oil and grease, we see the similarity. It was already confirmed that the properties of grease are primarily influenced by the base oil. The higher values for the grease were due to the effect of the thickener, which adheres to the surfaces and helps to build up the film [25], and also due to the greater quantity than in the case of the applied oil. The influence of these parameters was observed to be significant at speeds (1000–2000 mm/s) where centrifugal forces also have a greater impact. The experiment confirms that the centrifugal force has a positive effect on the layer thickness in the real configuration, although in the ball-on-disc configuration, starvation may occur [19]. It was found that the contact with a conformation of 0.52 showed more effective replenishment and higher thickness than the conformation of 0.58, when comparing the conformities. Conformity 0.52 produces a greater capillary effect, but the lubricant is pushed a further from the center of the track. Despite this greater extrusion, the average lubrication film values were higher than for the 0.58 conformity.

### 3.2. Volume of Lubricant around Contact at a Range of Speeds

Identical experiments were performed with grease while capturing the contact area. The contact itself and the rolling path behind the contact were captured. During the capturing of the path, an area of 56 mm$^2$ was recorded. This area corresponds to the size of the observation area shown in Figure 6. When recording the contact surrounding, the recorded region of interest was scaled according to the width of the contact and the height of the region was always 1.8 mm.

Figure 7 on the left shows three curves showing the evolution of the amount of lubricant around the contact with speed. The amount of lubricant was determined from the averaged magnitude of the fluorescence intensity in the region of interest. The measurement procedure and conditions were the same as in the previous section. The trends can be divided into two sections in the speed range 0–300 mm/s, when there was a significant decrease in the amount of lubricant. The second section was from 300–2000 mm/s and the values during this phase were more stable, with conformities of 0.58 and 0.52 showing an almost stable trend and conformity 0.55 showing a slightly linear decreasing trend. The difference between the conformities is mainly in the amount of lubricant around the contact, with conformity 0.58 having the highest volume of lubricant and conformity 0.52 having the smallest volume of lubricant. Figure 7 on the right shows three curves showing the evolution of the amount of lubricant on the rolling path. Instead of recording only the area around the contact, the entire area was captured as shown in Figure 8. The trends in the two graphs in Figure 7 are similar. It is possible to see an initial phase of 0–300 mm/s, where there is a more pronounced decrease, and a second phase of 300–2000 mm/s, where values remain stable. The conformity of 0.55 shows a slight decrease, as in the previous case. At conformity 0.52, significantly less lubricant was held on the rolling path than in the case of conformity 0.58.

Figure 8 compares the lubricant distribution for the selected conformities. Conformity 0.52 has less lubricant around the contact and the fingers on the rolling path are finer. For conformity 0.58, the amount of lubricant around the contact is greater and the fingers on the rolling path are thicker. It can be observed that the conformity has an effect on the location of the side strips on the rolling path.

The results show that the amount of lubricant around the contact and on the rolling path varies according to the conformities. Conformity 0.52 creates a smaller gap between the surfaces and thus there is less lubricant. When compared with Figure 4, it can be observed that conformity 0.52 is supplied with less lubricant and still creates better replenishment conditions. The most significant change was observed at low speeds when the input meniscus was approaching the contact. Thereafter, changes in the amount of lubricant were minimal. Although there was almost four times less lubricant in the vicinity of

0.52 conformity than in the case of 0.58 conformity, a thicker lubricant film was formed. The most significant difference between the conformities that aids film formation is the capillarity, which at a higher value will ensure film formation even with less lubricant. For longer bearing operation, when the film thickness decreases [26], a higher conformity may assist in a better lubricant replenishment process.

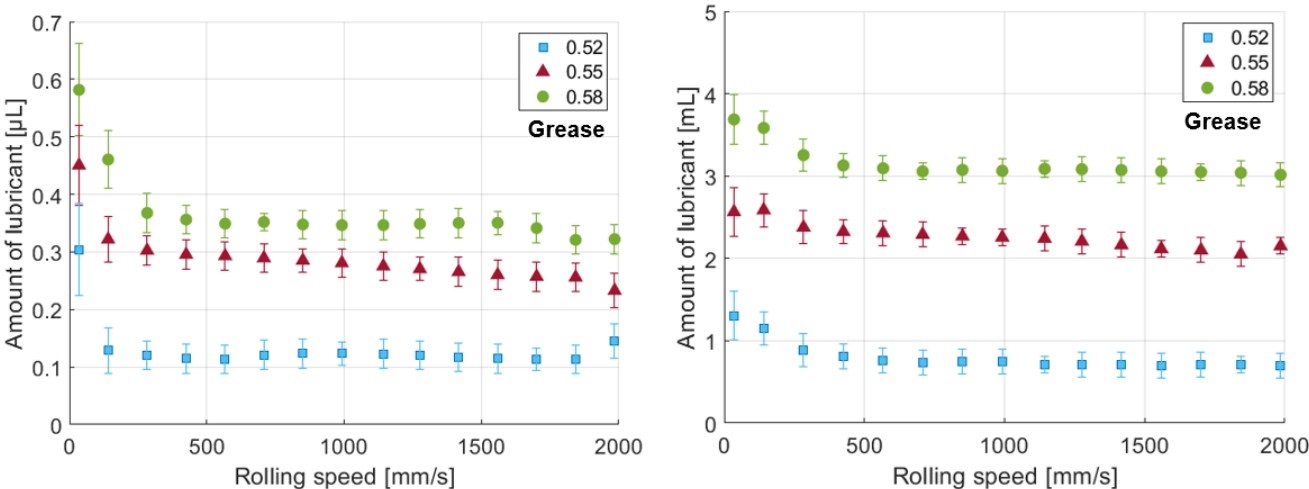

**Figure 7.** Amount of lubricant around the contact (**left**) and amount of lubricant on the rolling path (**right**).

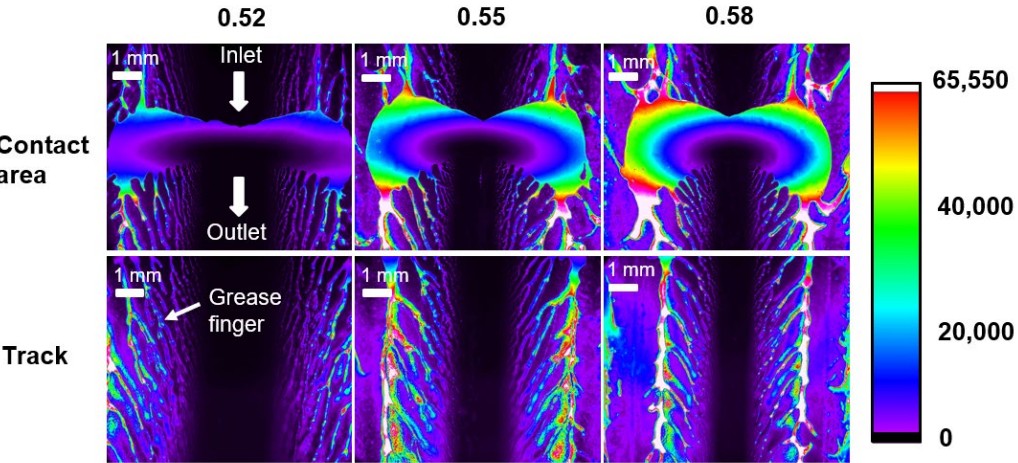

**Figure 8.** Grease distribution around the contact and on the rolling path.

### 3.3. Central Film Thickness at Constant Speed

Experiments were carried out with the base oil to determine the effect of the passes on the thickness of the contact film. A volume of 50 μL of oil was applied to one track in the same manner as in the initial experiments. The velocity was set at 500 mm/s and, after every 70 revolutions of the ring, a recording of the intensity from the contact was made.

Figure 9 on the left shows the results of this experiment with base oil, and the curves of the film thicknesses at the contacts have a decreasing trend for each conformity. During the initial rotations, the film thickness was the highest and corresponded to the thickness in Figure 4. There was a strong decrease in thickness in the 0–500 rpm range. From 500 rpm, the decrease was lower and almost unchanged from 1000 rpm. The difference between the conformities occurred along the entire length of the curve, with the 0.52 conformity showing higher film thickness values than the 0.58 conformity. The difference in central thickness at the end of the experiment was almost 15–20 nm and the difference in film thickness was in the range 45–60 nm.

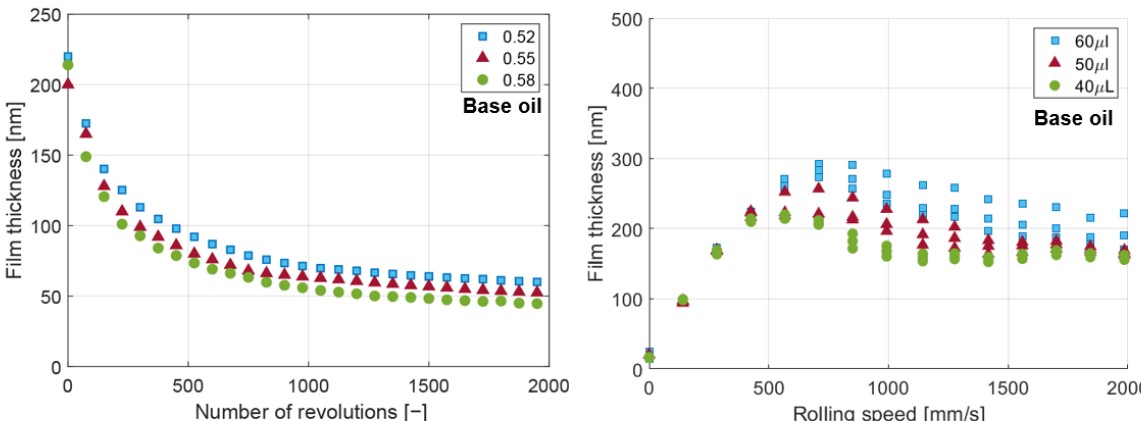

**Figure 9.** Effect of the number of revolutions on the development of thickness in contact (**left**). Influence of amount a lubricant (**right**) [conformity 0.52].

Figure 9 on the right shows the effect of the amount of lubricant (base oil) on the central film thickness in the range of the tested speed for a conformity of 0.52. Fully flooded conditions were observed up to 450 mm/s. The difference in film thickness occured only in the starvation phase, where a larger amount of applied oil also resulted in a larger film thickness. The values of film thicknesses were around 200 nm at the highest speed of 2000 mm/s.

When starvation conditions were created with a limited amount of lubricant, a gradual decrease was observed, as seen in Figure 9. This decrease was due to the gradual displacement of the lubricant, a process that has already been described by Venner et al. [26]. The differences in the conformities used (0.52–0.58) were quite small. The lubrication is only further assisted by a small amount of lubricant located on the raceway, which decreases due to the over-rollings. The main factor determining the degree of starvation is the available lubricant and speed. A conformity of 0.52 in all experiments showed the most efficient replenishment process, which resulted in the highest film thickness, while the least amount of lubricant was around this conformity. When different amounts of base oil were applied, a difference in film thickness was observed during starvation and a change in the speed at which starvation occurred.

## 4. Conclusions

The conclusion of this article can be summarized in a few points.

- It has been shown that different contact conformity shows different lubricant replenishment. The results showed that, under starvation conditions, a 0.52 conformity at speeds around 2 m/s produces a 50% thicker lubrication film than a 0.58 conformity. Better replenishment conditions were also observed in experiments with both base oil and grease.
- There was approximately three times less lubricant around a conformity of 0.52 than around a conformity of 0.58. A similar difference was also observed for lubricant on the rolling track. Despite this smaller amount of lubricant, during the experiments the smaller conformity had, on average, higher lubricating film thickness values than the larger conformity.
- Different conformities affect the hcc/hcff ratio. In grease lubrication, a conformity of 0.52, even at lower speeds, had a slightly lower value of this ratio than a conformity of 0.58. In contrast, at higher speeds, conformity 0.52 showed values around 0.4 and conformity 0.58 around 0.3. The average value of the hcc/hcff ratio was 0.68 for the 0.52 conformity and 0.61 for the 0.58 conformity.
- Experiments have confirmed that the lubricating properties of grease are significantly affected by the base oil used. In the case of the grease experiments, the film thickness was greater than that of the base oil.



**Author Contributions:** M.O. wrote this article. M.O. and D.K. designed and performed measurements and analyzed the results. P.S. and I.K. and M.H. supervised the work and provided suggestions for results discussion. All authors have read and agreed to the published version of the manuscript.

**Funding:** The research leading to these results has received funding from the Czech Science Foundation (grant No. 21-28352S).

**Conflicts of Interest:** The authors declare no conflict of interest.

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
