# Peer review of "Effect of Contact Conformity on Grease Lubrication"

_lubricants, doi:10.3390/lubricants10110289_

Round 1
Reviewer 1 Report
The manuscript gives a novel sight on the effect of conformity during lubricating. A minor revision is required, several concerns and questions should be well addressed before the manuscript is accepted.
1. In the apparatus, light would be reflected in the glass ring, how to calibrate the results to get the accurate size information (e.g. film thickness)?
2. Please provide specific definition of “hc” and “hcff” in the manuscript, and check out similar problems.
3. The very latest references about the lubricating mechanisms (e.g., ACS Appl. Mater. Interfaces 2021, 13, 55712−55725; Acta Materialia 232 (2022) 117934) are lacking, which may be helpful for introduction and/or discussion.
4. Point out the abstract definition in figures for better understanding, e.g., “the fingers on the rolling path” should be marked in Figure 8.
Author Response
Hello,
Thank you so much for your review of my article. Thanks to your revision, I managed to improve the existing manuscript again. The mentioned deficiencies have been incorporated into the manuscript and additional questions have been answered. The answers can be found in the attachment, so please see the attachment."

Reviewer 2 Report
The article considers the effect of contact conformity on grease lubrication. Data that may be of practical interest are presented. At the same time, unfortunately, the article is not without a number of shortcomings that must be eliminated before publication:
The Abstract contains a number of general arguments that do not provide new information and do not present the results of the article. I recommend starting the Abstract with "This work focuses...", omitting everything before that.
In general, the Abstract is too vague and non-specific. Too much attention is paid to the description of the methodology and too little to the results of the work. It is necessary to completely change the Abstract, focusing on the key results of the work, removing general arguments and an overly detailed description of the methodology (this can be done in the article itself, but not in the Abstract).
It is desirable to present specific (including quantitative) results in the Abstract: "more by ***%", "** times more accurate", etc.
The introduction also needs to be redone. The introduction is not simply an overview of the state of research in this field of knowledge. The task of the Introduction is to substantiate the goals and objectives of the work. It makes no sense to broaden the reader's horizons by giving him facts that are not directly related to the topic of research.
The format of the tables does not meet the requirements of the publisher.
There is a special character "±", do not use "+-"
Graphs (Fig. 4,5,7, etc.) - was only one experiment performed? Then the results cannot be considered sufficiently representative. If several experiments are carried out, error bars should be added.
I recommend making the conclusions more specific, including numerical values, not only qualitative assessments.
Author Response
Hello,
Thank you so much for your review of my article. Thanks to your revision, I managed to improve the existing manuscript again. The mentioned deficiencies have been incorporated into the manuscript and additional questions have been answered.